# Correlations between Sleep Features and Iron Status in Children with Neurodevelopmental Disorders: A Cross-Sectional Study

**DOI:** 10.3390/jcm12154949

**Published:** 2023-07-27

**Authors:** Donatella Giambersio, Lucia Marzulli, Lucia Margari, Emilia Matera, Lino Nobili, Elisa De Grandis, Ramona Cordani, Antonella Barbieri, Antonia Peschechera, Anna Margari, Maria Giuseppina Petruzzelli

**Affiliations:** 1Department of Neurosciences, Rehabilitation, Ophthalmology, Genetics, Maternal and Child Health (DINOGMI), University of Genoa, 16132 Genoa, Italy; giambersiodonatella@gmail.com (D.G.); elisadegrandis@gaslini.org (E.D.G.); ramona.cordani@libero.it (R.C.);; 2Department of Precision and Regenerative Medicine and Ionian Area (DIMEPRE-J), University of Studies of Bari “Aldo Moro”, 70124 Bari, Italy; 3Department of Translational Biomedicine and Neuroscience (DIBRAIN), University of Studies of Bari “Aldo Moro”, 70124 Bari, Italymaria.petruzzelli@uniba.it (M.G.P.); 4Child Neuropsychiatry Unit, IRCCS Istituto Giannina Gaslini, 16147 Genoa, Italy; 5Interdisciplinary Department of Medicine (DIM), University of Studies of Bari “Aldo Moro”, 70124 Bari, Italy

**Keywords:** sleep disturbances, neurodevelopmental disorders, autism spectrum disorder, attention deficit hyperactivity disorder, intellectual disability, children’s sleep habits questionnaire, ferritin, iron status biomarkers

## Abstract

A high prevalence of sleep disturbances has been reported in children with neurodevelopmental disorders (NDDs), such as autism spectrum disorder (ASD), attention deficit hyperactivity disorder (ADHD), and intellectual disability (ID). The etiology of sleep disorders in these children is heterogeneous and, recently, iron deficiency has received increasing attention. This study aims to investigate sleep features in children with NDDs and to explore a possible correlation between serum iron status biomarkers and qualitative features of sleep. We included 4- to 12-year-old children with a diagnosis of ASD, ADHD, or ID and assessed their sleep features through the children’s sleep habits questionnaire (CSHQ). Venous blood samples were collected to investigate ferritin, transferrin, and iron levels. The mean CSHQ total score exceeds the cut-off in all groups of children. In the ASD group, the Parasomnias subscale negatively correlated with serum ferritin levels (Rho = 0.354; *p* = 0.029). Our findings may suggest the existence of an association between iron status, sleep quality, and neurodevelopmental processes. In clinical practice, sleep assessment should be included in the routine assessment for patients with NDDs. Furthermore, a routine assessment of iron status biomarkers should be recommended for children with NDDs who have sleep disturbances.

## 1. Introduction

Sleep is a complex and fundamental neurophysiological process defined as a periodic suspension of the state of consciousness, associated with the slowing down of neurovegetative functions and the partial interruption of sensorimotor relations with the environment [1]. Sleep is regulated in its depth and intensity by a homeostatic process and by a circadian process that links the activity–rest cycle to environmental stimuli, such as the alternation of light and dark. This latter process is controlled by the suprachiasmatic nucleus, which activates the production of melatonin by the pineal gland. Disrupted sleep can negatively impact the cardiovascular, immune and metabolic systems, including growth disorders. Sleep deprivation also impairs performance on motor and cognitive tasks [2], whereas sleep enhances cognitive functions such as visual discrimination [3], motor learning [4], and insight [5], which underlie the learning mechanism [4,5,6,7]. It has been demonstrated that poor sleep hygiene could have negative impacts on children’s attention span, mood regulation and behaviour. For these reasons, sleep disturbances in children are associated with behavioral issues and impaired cognitive performance, and they are a major concern in children with neurodevelopmental disorders (NDDs) [8]. Sleep disorders in children also significantly affect the health and wellbeing of the child, but also of other family members with an increase of parenting stress.

Indeed, up to three-quarters of children and adolescents with NDDs have a sleep problem [9,10], which tends to persist over time [11]. The etiology of sleep disorders in children with NDDs is largely heterogeneous, with contributing factors including environmental influences, caregiver-related factors, and disease-related etiologies Patients with NDDs, such as Autism Spectrum Disorder (ASD), Attention Deficit Hyperactivity Disorder (ADHD) and Intellectual Disability (ID), often experience sleep disturbances that affect clinical presentation, daily functioning and quality of life. Sleep alteration during developmental age may be considered as directly related to atypical brain development in neurodevelopmental disorders, as well as an additional risk factor for cognitive and behavioral impairment. Understanding the specific characteristics of sleep disturbances in patients with NDDs is critical to defining management strategies that can be of benefit to both the child and the family [12]. In recent years, iron deficiency in people with sleep disorders has received increasing attention. Iron deficiency has been shown to be associated with restless legs syndrome [13], arousal and respiratory sleep disorders [14,15]. The mechanism behind this association may be related to the central role of iron as a co-factor in neurotransmitter synthesis, myelination, and oxygen transport. Moreover, the sleep–wake cycle is at least partly controlled by the dopamine–opiate system, in which iron is an essential cofactor, so its deficiency could lead to damage to brain structures involved in sleep, such as the hippocampus or basal ganglia [16,17]. On the other hand, given the central role of iron in the brain as a cofactor in dopamine and serotonin synthesis, as well as in ATP production and myelination, iron deficiency has been positively associated with the risk of developing NDDs, and there is also evidence that postnatal iron deficiency may exacerbate pre-existing NDDs [18]. Although there are studies investigating the association between iron status and sleep disturbances in both typically developing children and children with NDDs, to our knowledge there are no studies directly comparing iron status in children with different NDDs [19,20,21,22]. Considering such premises, the aims of this study were (1) to compare the qualitative features of sleep disturbances in a sample of children with ASD, ADHD and ID through the administration of the children’s sleep habits questionnaire (CSHQ); (2) to confront the serum iron status biomarkers between children with ASD, ADHD and ID; (3) to identify a possible correlation between serum iron status biomarkers and qualitative features of sleep in these three subgroups of children.

## 2. Materials and Methods

### 2.1. Study Population

This was an observational cross-sectional study involving children aged between 4 and 12 years consecutively referred to the child neuropsychiatric units of the University Hospital of Bari and the “Giannina Gaslini” Institute of Genoa between January and October 2022. Inclusion criteria for enrolled patients were a diagnosis of ASD, ADHD or ID according to the Diagnostic and Statistical Manual for Mental Disorders, Fifth Edition (DSM-5) [23], based on clinical judgment by an expert team of child and adolescent neuropsychiatrists and psychologists, and score profiles from standardized diagnostic tools including the Conners’ Parent Rating Scale, Revised (CPRS-R) [24], Swanson, Nolan and Pelham Rating Scale (SNAP-IV) [25], Autism Diagnostic Observational Schedule, Second Edition (ADOS-2) [26,27], Wechsler Preschool and Primary Scale of Intelligence-Third Edition (WPPSI-III) [28], Wechsler Intelligence Scales for Children-Fourth Edition (WISC-IV) [29], Leiter International Performance Scale-Revised (Leiter-R) [30], and Vineland Adaptive Behavior Scale (VABS) [31]. Exclusion criteria were the presence of diseases that could affect sleep (e.g., multiple sclerosis, headache, atopic dermatitis, food intolerance), diseases that could alter iron status (e.g., liver and autoimmune diseases, hemochromatosis, acute and chronic infections), and the use of hypnotic–sedative drugs (e.g., benzodiazepines). Informed consent was obtained from parents/guardians before enrolment. The study was approved by the Interregional Ethics Committee of the University Hospital of Bari (protocol number 0085949|07/10/2022).

### 2.2. Assessment of Sleep Disturbances

Subjective/qualitative features of sleep were evaluated through the administration of the children’s sleep habits questionnaire (CSHQ) filled out by the patients’ caregivers. The CSHQ is a retrospective, 33-item questionnaire consisting of eight subscales (bedtime resistance, sleep onset delay, sleep duration, sleep anxiety, night waking, sleep-disordered breathing (SDB), daytime sleepiness, and parasomnias) with items assessing clinical symptoms of sleep disorders according to the International Classification of Sleep Disorders in Children [32]. Parents are asked to recall sleep behaviors that occurred during a ‘typical’ recent week. Each item is rated on a three-point scale: ‘usually’ if the sleep behavior occurred five to seven times/week; ‘sometimes’ for two to four times/week; and ‘rarely’ for zero to one time/week. A CSHQ total score of 41 has been reported as a sensitive clinical cut-off for identifying probable sleep problems. The translation of the questionnaire into Italian showed that the CSHQ-IT is equivalent to the original English version [33], making it not only a test with high sensitivity and specificity but also a valid sleep screening measure for research purposes. We defined “good sleepers” as children who did not exceed the total cut-off score of 41 on the CSHQ, and “poor sleepers” as children who exceeded the total cut-off score of 41 on the CSHQ.

### 2.3. Assessment of Iron Status Biomarkers

Venous blood samples were collected from all patients between 8:00 and 9:00 am after an overnight fast. All patients remained in the supine position during sampling to avoid prick exposure. Serum levels of ferritin, iron and transferrin were measured in the clinical pathology laboratory of the University Hospital of Bari. The serum was extracted from the centrifugation of blood and obtained after the clotting of blood. Ferritin was measured using the immunoassay with direct chemiluminometric methodology. The intra-assay coefficient of variation (CV) was <7% with a range between 4.0% and 6.3%; the detection limit was ≤1.0 ng/mL (2.2 pmol/L) and the reference values (95th percentile) were 22–322 ng/mL for males, and 10–291 ng/mL for females. Iron was measured by the colorimetric method, with inter-assay CVs <3%, which ranged from 0.4% to 2.6%. The detection limit was ≤10 µg/dL (≤1.8 µmoL/L) and reference values were 65–175 µg/mL for males (11.6–31.3 µmol/L) and 50–170 µg/mL (9.0–30.4 µmol/L) for females. Transferrin was measured by the immunonephelometric method. Intra-assay CVs were <3.5% (range 1.99–3.46%). The limit of quantification was 8.75 mg/dL [0.09 g/L] and reference values ranged from 200 to 360 mg/dL [2.00–3.60 g/L].

### 2.4. Statistical Analysis

Demographic data were presented as mean and standard deviations (SD) for continuous variables and as frequencies and percentages for categorical variables. Comparisons of quantitative variables were made between the three subgroups using ANOVA or Kruskal–Wallis for independent samples, appropriately selected according to the assessment of normality assessed by the Shapiro–Wilk test. Pearson’s or Spearman’s correlation tests, appropriately selected according to the assessment of normality, were used to examine the relationship between iron status biomarkers and the CSHQ total score and subscales scores. Jamovi software version 2.3.17 was used for statistical analysis. A level of *p* < 0.05 indicated statistical significance.

## 3. Results

### 3.1. Sociodemographic and Clinical Features of the Sample

The sample consisted of 106 children, 38 with a diagnosis of ASD (30 males, 8 females; mean age 6.79 ± 2.02), 49 with a diagnosis of ADHD (33 males, 16 females; mean age 8.77 ± 2.04), and 19 with a diagnosis of ID (11 males, 8 females; mean age 7.22 ± 2.99).

The mean CSHQ total score exceeds the specific cut-off in all groups of children, that is to say in each group the score was greater than 41. The comparison of mean CSHQ total score between the three groups, assessed using the Kruskal–Wallis test, showed no significant statistical differences. Moreover, in all the groups there was a high percentage of children defined as “poor sleepers”: ASD group had 78.95%, the ADHD group 75.51% and the ID group 73.68% of poor sleepers.

Regarding the qualitative characteristics of sleep, no significant statistical differences were found between the groups in the comparison of CSHQ subscales’ scores (Table 1).

The mean values of serum biomarkers of iron status (ferritin, transferrin, and iron) were not below the lower reference value in the three subgroups. Through the comparison of iron status indicators between the three groups, it was found that the ID group had the lowest mean values for ferritin (37.2 ± 22.7 ng/mL) and transferrin (265 ± 42.0 mg/dL), while the ASD group had the lowest mean values for serum iron (77.1 ± 28.2 µg/dL), but none of these comparisons reached statistical significance (Table 2).

### 3.2. Correlation between Iron Status Biomarkers and CSHQ Total Score and Subscales

Correlation analysis between iron status biomarkers (ferritin, transferrin, and serum iron) and CSHQ total score and subscales (Bedtime resistance, Sleep onset delay, Sleep duration, Sleep anxiety, Night wakings, Parasomnias, SDB, Daytime sleepiness) in the three subgroups is shown in Table 3, Table 4 and Table 5 and in Figure 1. In the ASD group, serum ferritin levels were negatively correlated with the Parasomnias subscale (Spearman’s Rho coefficient of correlation: −0.354) with a statistical significance (*p* = 0.029). None of these correlations reached statistical significance in the ADHD and ID subgroups.

## 4. Discussion

In accordance with previous literature [8,34,35], our study confirms that patients of developmental age with a diagnosis of NDDs have a high risk of suffering from sleep disruptions. The observed prevalence of sleep abnormalities above 70% in all three studied subgroups exceeds prevalence estimates in the general age-matched population [36]. This poorer sleep quality in children with NDDs compared to typically developing children may suggest the existence of an association between sleep and neurodevelopmental processes. However, it is not clear whether the association is transversely identifiable in the various neurodevelopmental disorders or if instead distinctive patterns of interaction between sleep and neurodevelopment can be identified in different NDDs. A possible bi-univocal association between sleep disturbance and NDDs or a common etiopathogenesis could be suggested. On the one hand, sleep disturbances may contribute to the exacerbation of brain developmental changes typical of NDDs [37], or may significantly affect the severity of NDD symptoms and exacerbate daytime behavioral disturbances such as aggression, irritability, inattention and hyperactivity [38]; on the other hand, NDDs symptoms and their associated psychiatric and organic comorbidities, such as anxiety, depression, epilepsy, recurrent pain, cough, constipation, may increase the severity of sleep disturbance, creating a vicious cycle [39]. For example, poor sleep in ASD may lead to disturbances in neurotransmitters, including melatonin and serotonin. Genetic studies have shown that polymorphisms in circadian CLOCK genes and alterations in the NLGN/NRXN/SHANK3 synaptic pathway may alter the clock and circadian rhythms in individuals with ASD [40]. However, in children with ASD, sensory integration deficits, ritualized or self-injurious behaviors, poor communication skills, and limited responsiveness to social cues may also exacerbate bedtime resistance, prolonging the time between turning off the lights and sleep onset, and thus disrupting sleep continuity. Regarding ADHD, a growing body of data suggests that ADHD and sleep disorders share a common neurological etiology [41] involving the catecholaminergic system and brain regions such as the dorsolateral and ventrolateral prefrontal cortex, the dorsal anterior cingulate cortex, and certain regions of the thalamus (responsible for sleep spindles and synchronization) [42]. Children with ID may have difficulty registering or understanding environmental cues about when it is time to sleep due to an alteration in the brain systems involved in sleep control.

In this study, we did not find significant differences in the total scores and subscales scores of the CSHQ of the three examined subgroups. To our knowledge, this is the first study that directly compares and correlates sleep quality features and iron status in children with ASD, ADHD, and ID, while data are mainly available for the individual diagnostic groups [21,38,43,44,45,46,47,48,49,50,51,52,53,54], so further studies are needed to confirm our results.

The second finding is the significant negative correlation between mean serum ferritin levels and CSHQ Parasomnias subscale scores in the ASD group. As well hypothesizing a bidirectional clinical and etiopathogenetic relationship between NDDs, including autism, and sleep disorders, we also see evidence to support the existence of a similar relationship between serum ferritin levels and ASD. Ferritin is the most widely used biomarker to study iron deficiency in the human body. Its decrease, in the absence of hyposideraemia and normal transferrin levels, indicates the initial depletion of iron stores [55]. Iron is an essential component of several proteins and enzymes that perform important functions in metabolism and survival, including protecting the genome from damage and mutation. In the brain, it is involved in white matter myelination and the function of neurotransmitters such as dopamine, norepinephrine, and serotonin. Iron deficiency could be the result of taste hypersensitivity, which is more common in preschool children with autism [56]. Low serum iron levels may also be associated with more severe autistic symptoms, although there are conflicting data in the literature [56]. Given the role of iron in regulating dopaminergic function, which affects the sleep–wake cycle [57], its deficiency has been shown to be associated with several sleep disorders. Parasomnias are undesirable events that occur during sleep transition, REM sleep, or the transition from N3 sleep to lighter N2/N1 sleep [58]. Activation of central pattern generators in the brainstem and spinal cord, which can trigger parasomnias, can be produced by sensory stimuli from the upper airways (as in obstructive sleep apnea), gastrointestinal tract (as in gastro-esophageal reflux) and extremities (as in restless legs syndrome) [58]. It has been shown that restless legs syndrome is often associated with low serum ferritin levels in children with ASD [22,59]. Therefore, our results might lead us to hypothesize that, in our ASD children with parasomnias, patients with low serum ferritin levels might have restless legs syndrome, which is the trigger for the development of such a sleep disorder. However, to our knowledge, in the literature, there are no data yet linking sleep disturbances and iron status in ASD children.

This study has some limitations, including the lack of a typically developing control group and the small sample size. Therefore, further studies on larger samples are needed to allow the generalizability of our results. In addition, sleep was only assessed using a subjective measure, the CSHQ, which has the advantage of saving time and money and measuring a wide range of sleep parameters, but a further objective sleep assessment with polysomnography could add important complementary findings.

## 5. Conclusions

This study underlines the existence of a link between sleep disturbances and NDDs. Future research on this topic could provide a deeper understanding of the relationship between atypical brain networks and NDD phenotypes, helping to clarify how they may cause, exacerbate, or result from poor sleep. In addition, our findings highlight an association between sleep disturbances, such as parasomnias, and iron deficiency in ASD patients. Although further studies are needed to confirm these results, we recommend routine assessment of iron status biomarkers in children with NDDs who have sleep disturbances. Future studies should clarify the role of iron supplementation in children with low serum ferritin levels and sleep disturbances, before or in addition to traditional therapy [14,22,60]. Finally, considering the negative impact of sleep disorders on neurodevelopment, it would be desirable to conduct studies that correlate the severity of NDD, symptoms in children with sleep disorders and iron biomarkers, possibly measuring additional indicators of iron status.

Several categories of iron metabolism markers have already been measured for prognostic purposes in children with febrile urinary tract infection and in children after intensive chemotherapy for acute leukemias or undergoing hematopoietic cell transplantation. So, parameters determining functional and storage iron pool (iron, total iron-binding capacity, transferrin, ferritin, ferritin heavy and light chain), proteins contributing to the absorption of iron and its release from tissues stores (hepcidin, soluble ferroportin-1 and soluble hemojuvelin) and proteins determining the erythropoietic activity of the bone marrow (erythropoietin, erythroferrone, and soluble transferrin receptor) [61,62] could be considered for future research also in this area.

## Figures and Tables

**Figure 1 jcm-12-04949-f001:**
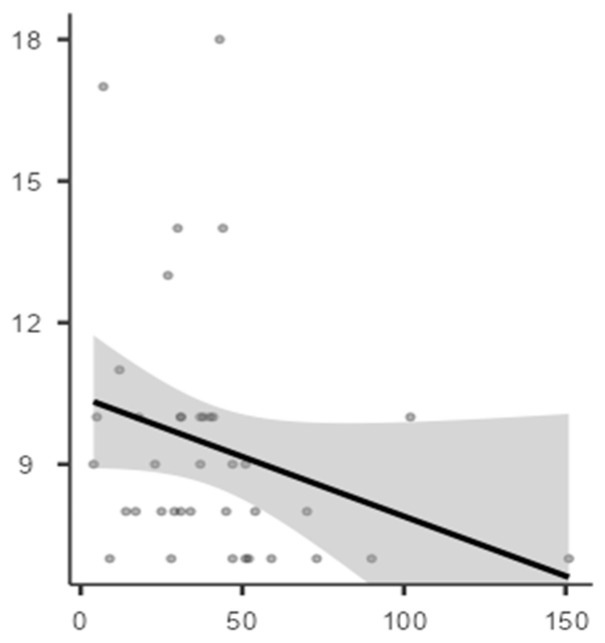
Correlation between serum ferritin levels (X axis) and Parasomnias (Y axis) in ASD group.

**Table 1 jcm-12-04949-t001:** Comparison of CSHQ total and subscales score (mean ± standard deviation) in ASD, ADHD, and ID children (Kruskal–Wallis test).

	ASD (*n* = 38)	ADHD (*n* = 49)	ID (*n* = 19)	*p*-Value
CSHQ total score	49.2 ± 10.3	49.4 ± 9.53	48.4 ± 9.77	0.537
Bedtime resistance	11.1 ± 3.52	10.2 ± 3.55	10.4 ± 3.44	0.179
Sleep onset delay	1.47 ± 0.725	1.55 ± 0.738	1.32 ± 0.671	0.337
Sleep duration	3.89 ± 1.59	4.11 ± 1.75	3.53 ± 0.905	0.642
Sleep anxiety	6.93 ± 2.60	7.13 ± 2.67	6.58 ± 2.01	0.326
Night wakings	4.26 ± 1.50	4.45 ± 1.72	4.16 ± 1.54	0.939
Sleep-disordered breathing	3.91 ± 1.38	3.84 ± 1.41	3.86 ± 1.21	0.855
Daytime sleepiness	12.6 ± 3.39	11.8 ± 2.54	13.1 ± 3.75	0.307
Parasomnias	9.35 ± 2.31	9.39 ± 2.67	9.05 ± 2.72	0.392

**Table 2 jcm-12-04949-t002:** Mean values of serum iron status biomarkers in ASD, ADHD and ID children (Kruskal–Wallis test was used for ferritin and iron, ANOVA was used for transferrin).

	ASD (*n* = 38)	ADHD (*n* = 49)	ID (*n* = 19)	*p*-Value
Ferritin (mean ±SD)	40.7 ± 28.5 ng/mL	43.6 ± 33.0 ng/mL	37.2 ± 22.7 ng/mL	0.727
Transferrin (mean ± SD)	270 ± 45.5 mg/dL	279 ± 35.2 mg/dL	265 ± 42.0 mg/dL	0.356
Iron (mean ± SD)	77.1 ± 28.2 µg/dL	84.0 ± 33.6 µg/dL	79.7 ± 26.2 µg/dL	0.832

**Table 3 jcm-12-04949-t003:** Correlation between iron status biomarkers and CSHQ total score and subscales in the ASD group (Spearman’s correlation test was used for ferritin; Pearson’s correlation test was used for transferrin and iron).

		CSHQ Total	Bedtime Resistance	Sleep Onset Delay	Sleep Duration	Sleep Anxiety	Night Wakings	Parasomnias	SDB	Daytime Sleepiness
Ferritin (ng/mL)	Rho *p-*value	−0.173 0.299	0.012 0.942	−0.063 0.706	−0.088 0.601	−0.032 0.851	0.007 0.967	−0.354 0.029 *	−0.012 0.945	−0.179 0.281
Transferrin (mg/dL)	r *p*-value	0.123 0.797	0.043 0.797	0.121 0.469	0.090 0.589	0.127 0.448	−0.056 0.740	0.022 0.897	0.045 0.790	0.265 0.109
Iron (µg/dL)	r *p*-value	0.198 0.233	0.198 0.235	0.098 0.056	0.209 0.209	0.108 0.518	0.143 0.392	0.177 0.288	0.077 0.647	−0.022 0.894

* significant correlation; SDB: sleep-disordered breathing.

**Table 4 jcm-12-04949-t004:** Correlation between iron status biomarkers and CSHQ total score and subscales in the ADHD group (Spearman’s correlation test was used for ferritin, and iron; Pearson’s correlation test was used for transferrin).

		CSHQ Total	Bedtime Resistance	Sleep Onset Delay	Sleep Duration	Sleep Anxiety	Night Wakings	Parasomnias	SDB	Daytime Sleepiness
Ferritin (ng/mL)	Rho *p-*value	0.114 0.435	0.132 0.365	0.036 0.806	−0.088 0.547	0.244 0.091	0.099 0.498	0.048 0.743	−0.044 0.766	0.081 0.580
Transferrin (mg/dL)	r *p-*value	0.126 0.389	0.005 0.972	−0.057 0.699	−0.025 0.862	0.100 0.495	0.077 0.601	−0.044 0.763	0.141 0.333	0.181 0.214
Iron (µg/dL)	Rho *p-*value	0.017 0.908	0.049 0.738	0.115 0.430	0.042 0.774	0.121 0.407	−0.201 0.167	−0.048 0.746	0.222 0.125	0.010 0.943

SDB: sleep-disordered breathing.

**Table 5 jcm-12-04949-t005:** Correlation between iron status biomarkers and CSHQ total score and subscales in the ID group (Spearman’s correlation test was used for ferritin; Pearson’s correlation test was used for transferrin and iron).

		CSHQ Total	Bedtime Resistance	Sleep Onset Delay	Sleep Duration	Sleep Anxiety	Night Wakings	Parasomnias	SDB	Daytime Sleepiness
Ferritin (ng/mL)	Rho *p-*value	−0.059 0.812	−0.436 0.062	0.433 0.064	0.200 0.412	−0.266 0.272	0.060 0.808	−0.141 0.565	−0.053 0.830	0.195 0.424
Transferrin (mg/dL)	r *p-*value	−0.193 0.428	0.225 0.354	−0.406 0.085	−0.253 0.295	0.055 0.824	−0.164 0.501	−0.157 0.522	−0.249 0.303	−0.257 0.287
Iron (µg/dL)	r *p-*value	−0.022 0.927	−0.149 0.543	−0.316 0.188	−0.280 0.245	0.094 0.701	0.167 0.495	−0.068 0.784	0.002 0.992	0.191 0.435

SDB: sleep-disordered breathing.

## Data Availability

The data presented in this study are available on request from the corresponding author.

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
