# Peer review of "Correlations between Sleep Features and Iron Status in Children with Neurodevelopmental Disorders: A Cross-Sectional Study"

_jcm, 2023, doi:10.3390/jcm12154949_

Round 1
Reviewer 1 Report
The topic of searching for the etiopathology of sleep disorders in children, raised by the authors, is an extremely important clinical problem in pediatric neurology. Therefore, I consider the manuscript to be a valuable observation and will consider accepting it after taking into account all my comments on the text of the paper.
Here are the comments on the manuscript:
1. title: A cross-sectional study is by design an observational study, therefore I suggest deleting the word "observational".
2. abstract: it is not necessary to capitalize the names of diseases. The authors should add a correlation coefficient to the p-value.
3. keywords: please do not use abbreviations.
4. the introduction is well-written and the purpose of the research is clearly defined.
5. methods: more data, how the levels of the tested iron metabolism parameters were measured - what methods were used, what are their CVs, detection limits, reference values. How was the blood serum obtained?
6. results - table 3 is illegible. In addition, the results, although pilot and strongly limited by poor statistical analysis, are clearly presented in fact.
7. Discussion - please discuss whether other laboratory parameters of iron metabolism can be used in further research by the authors. Please discuss this aspect in light of the latest publications on iron metabolism in children: Cancers (Basel). 2023;15(4):1041 and Children (Basel). 2023;10(5):870.
Author Response
Dear reviewer,
we sincerely thank you for your suggestions. We hope to be able to fully satisfy your requests in order to improve the scientific quality of our manuscript. Here below you will find our responses to your comments.
1.title: A cross-sectional study is by design an observational study, therefore I suggest deleting the word "observational".
Following your suggestion, we deleted the word “observational”.
- abstract: it is not necessary to capitalize the names of diseases. The authors should add a correlation coefficient to the p-value.
We eliminated capital letters for the diseases names and added the correlation coefficient to the p-value.
- keywords: please do not use abbreviations.
We eliminated abbreviations in the keywords section.
- the introduction is well-written and the purpose of the research is clearly defined.
We are grateful to you for this consideration.
- methods: more data, how the levels of the tested iron metabolism parameters were measured - what methods were used, what are their CVs, detection limits, reference values. How was the blood serum obtained?
As you suggested, we included the methods used to measure the iron metabolism parameters, the CVs, the detection limits, the reference values and the process by which the serum was obtained.
- results - table 3 is illegible. In addition, the results, although pilot and strongly limited by poor statistical analysis, are clearly presented in fact.
We are sorry for the error in table 3 which has been corrected.
- Discussion - please discuss whether other laboratory parameters of iron metabolism can be used in further research by the authors. Please discuss this aspect in light of the latest publications on iron metabolism in children: Cancers (Basel). 2023;15(4):1041 and Children (Basel). 2023;10(5):870.
As you rightly suggested, we included other laboratory parameters of iron metabolism that could be used in further research considering the latest publications on iron metabolism.
Best regards,
Emilia Matera & Lucia Margari
Reviewer 2 Report
The authors have explored a very important topic, as evidenced by the carefully prepared introduction. Here are some minor comments:
1. A statistically significant correlation (serum ferritin levels with the Parasomnias) should be plotted.
2. I suggest you add the units in Table 2.
3. Minor editorial corrections should be made (for example: µg and not ug, ANOVA and not Anova)
The discussion was carried out in a very thorough way, taking into account many references.
Author Response
Dear reviewer,
we really appreciate your suggestions to improve the quality of this manuscript. Here below we list the responses to your comments; we hope you find them satisfactory.
1.A statistically significant correlation (serum ferritin levels with the Parasomnias) should be plotted.
As properly suggested, we added a figure to show the significant correlation relationship between ferritin and parasomnias in the ASD group (figure 1).
- I suggest you add the units in Table 2.
We have done this correction.
- Minor editorial corrections should be made (for example: µg and not ug, ANOVA and not Anova)
As you suggested in points 1, 2 and 3, we plotted the statistically significant correlation (serum ferritin levels with the Parasomnias), and did minor editorial corrections in the Results section.
The discussion was carried out in a very thorough way, taking into account many references.
We are really grateful for your positive comment.
Best regards,
Emilia Matera & Lucia Margari
Round 2
Reviewer 1 Report
The authors have addressed all my concerns.